# *R2R3-MYB* Gene Family in *Coptis teeta* Wall.: Genome-Wide Identification, Phylogeny, Evolutionary Expansion, and Expression Analyses during Floral Development

**DOI:** 10.3390/ijms25168902

**Published:** 2024-08-15

**Authors:** Jichen Yu, Shaofeng Duan, Zhenyang Shua, Kecheng Li, Guisheng Xiang, Timothy Charles Baldwin, Yingchun Lu, Yanli Liang

**Affiliations:** 1The Key Laboratory of Medicinal Plant Biology of Yunnan Province, National-Local Joint Engineering Research Center on Germplasm Innovation, Utilization of Chinese Medicinal Materials in Southwest, College of Agronomy and Biotechnology, Yunnan Agricultural University, Kunming 650201, China; yujcstart@163.com (J.Y.); likecheng0610@163.com (K.L.); xianggs1981@163.com (G.X.); 2National Key Laboratory of Wheat and Maize Crop Science, College of Life Sciences, Henan Agricultural University, Zhengzhou 450002, China; 3Faculty of Science and Engineering, University of Wolverhampton, Wolverhampton WV1 1LY, UK; t.baldwin@wlv.ac.uk; 4College of Education and Vocational Education, Yunnan Agricultural University, Kunming 650201, China; 18203535541@163.com

**Keywords:** *Coptis teeta*, *R2R3-MYB* gene family, evolutionary analysis, dichogamy, herkogamy, floral development, *Ranales*

## Abstract

The *R2R3-MYB* gene family represents a widely distributed class of plant transcription factors. This gene family plays an important role in many aspects of plant growth and development. However, the characterization of *R2R3-MYB* genes present in the genome of *Coptis teeta* has not been reported. Here, we describe the bioinformatic identification and characterization of 88 *R2R3-MYB* genes in this species, and the identification of members of the *R2R3-MYB* gene family in species within the order *Ranales* most closely related to *Coptis teeta*. The *CteR2R3-MYB* genes were shown to exhibit a higher degree of conservation compared to those of *A. thaliana*, as evidenced by phylogeny, conserved motifs, gene structure, and replication event analyses. Cis-acting element analysis confirmed the involvement of *CteR2R3-MYB* genes in a variety of developmental processes, including growth, cell differentiation, and reproduction mediated by hormone synthesis. In addition, through homology comparisons with the equivalent gene family in *A. thaliana*, protein regulatory network prediction and transcriptome data analysis of floral organs across three time periods of flower development, 17 candidate genes were shown to exhibit biased expression in two floral phenotypes of *C. teeta*. This suggests their potential involvement in floral development (anther development) in this species.

## 1. Introduction

Plant-associated *MYB* transcription factors represent one of the largest and most diverse transcription factor families. MYB proteins contain a DNA-binding region called the MYB domain which consists of approximately 50 amino acids and is highly conserved within the family [1]. The diversity of the *MYB* gene family in plants is believed to have arisen through gene duplication and divergent evolution, resulting in specific and highly conserved functions in various plant species and organs. Among them, the *R2R3-MYB* gene family consists of two tandem repeats of the DNA-binding domain of MYB (R2 and R3), separated by a short linker region. Previous research revealed that the *R2R3-MYB* gene family plays a crucial role in plant growth and development [1]. The *R2R3-MYB* gene family of several plant species has been studied, indicating widespread involvement of family members in various aspects of plant growth and development [2,3,4]. These include secondary metabolism, cell differentiation, trichome development, root hair formation, and abiotic stress responses [5,6,7].

Recent studies have shown that members of the *R2R3-MYB* transcription factor family participate in the regulation of photomorphogenesis [8]. *MYB112* enhances the transcriptional activation of PHYTOCHROME-INTERACTING FACTOR 4 (*PIF4*), which fosters the expression of hormone-related genes. This augmentation boosts the biosynthesis of plant growth regulators and signaling which regulate hypocotyl growth within the circadian rhythm. The *R2R3-MYB* gene family also has an important role in secondary cell wall biosynthesis. Three *R2R3-PlMYB* transcription factors (*PlMYB43*, *PlMYB83*, and *PlMYB103*) have been shown to promote lignin deposition and secondary cell wall thickening, thereby enhancing the strength of floral stems [9]. In addition, some members of R2R3-MYB gene family can regulate petal color formation [10,11]. In a study of anthocyanin biosynthesis in *Petunia*, two highly expressed MYB-type transcription factors, along with members of the WD40 and bHLH gene families, were shown to collectively regulate the expression of genes involved in anthocyanin synthesis, thereby affecting petal color formation [12].

It has been reported that the *R2R3-MYB* gene family can influence the morphology and function of flowers by regulating the development of floral organs such as petals, stamens, and carpels [13,14,15]. Research on floral sex determination showed that in tomato, the gene *MYB21* regulates the expression of genes related to jasmonic acid (JA) synthesis, activating the auxin and gibberellin signaling pathways, thus influencing carpel development [16]. In *Arabidopsis*, numerous *R2R3-MYB* transcription factors regulate pollen development, while in *Marchantia*, defects in the female-specific gene *MpFGMYB* result in the transformation of female to male gender [17,18]. The transcription factor *MYB108* plays a crucial role in regulating plant reproduction. Silencing this gene through RNA interference techniques or gene knockout methods can significantly delay anther dehiscence and reduce pollen viability [19]. This phenomenon indicates that *MYB108* is involved in the regulation of pollen maturation and anther dehiscence. Due to the reduction in pollen viability and the delay in anther dehiscence, the rate of self-fertilization in these plants is significantly decreased, thereby promoting outcrossing and enhancing genetic diversity [3].

*Coptis teeta* Wall., a monoecious plant widely used in Traditional Chinese Medicine (TCM), displays both dichogamous and herkogamous reproductive strategies [20]. This species can bear maternal flowers (type I) that are protogynous with exserted stigmas, and paternal flowers (type II) that are protandrous with concealed stigmas. The traits of female precocity and forward herkogamy are interconnected, as are those of male precocity and reverse herkogamy, but the molecular mechanism of these traits in *C. teeta* is currently poorly understood. Since the *R2R3-MYB* gene family is known to play a crucial role in floral organ development, we identified 88 R2R3-MYB proteins in *C. teeta* and deciphered their structure, protein motifs, and evolutionary relationships in the *Ranunculaceae*. We also identified 17 genes that have biased expression between Type I and Type II flowers, suggesting that these genes may be involved in dichogamy and herkogamy in *C. teeta.*

## 2. Results

### 2.1. Identification and Bioinformatic Analyses of the R2R3-MYB Gene Family in C. teeta

The *AtR2R3-MYB* gene family was retrieved from the TAIR database (https://www.arabidopsis.org/, accessed on 20 February 2023), and these genes were then employed as reference sequences used to perform BLASTP searches. The HMM file of the MYB DNA-binding domain (PF00249) was acquired from the Pfam database (http://pfam.xfam.org/, accessed on 20 February 2023) to facilitate the identification of candidate *CteR2R3-MYB* genes. To merge the two results, the candidates were subjected to cross-validation utilizing SMART and NCBI-CDD. As a result of this, eighty-eight R2R3-MYB proteins were identified using high-quality and complete chromosome genome data (Appendix A) in the *C. teeta* genome. The isoelectric point of the predicted R2R3-MYB proteins ranged from 4.74 to 9.75, and the predicted molecular weights ranged from 15,660.04 Da to 231,532.32 Da. Protein secondary structure prediction indicated a predominance of α-helices and disordered coils in all the predicted R2R3-MYB proteins, followed by extended strands and β-turns. In addition, subcellular localization predictions for the R2R3-MYB proteins showed that 87 of them were predicted to be located in the nucleus, with one (*CteMYB11*) in the chloroplast (Appendix A).

In order to obtain more information, we analyzed the conserved motifs, gene structures, and the evolutionary relationships of the members of this gene family (Appendix A). We predicted 10 motifs within the *CteR2R3-MYBs* using the online MEME program (Appendix A). Nearly 89% of these genes contained motifs 1, 2, 3, and 4, indicating significant levels of conservation and functional importance. In addition, motifs 5 and 6 were also conserved, while motifs 9 and 10 were found in only a few genes. Notably, motifs 7 and 8 co-occurred within the same cluster of *R2R3-MYB* genes. Analysis of the gene structure revealed that the number of exons ranged from 2 to 16 and the number of introns from 1 to 15 (Appendix A). It is noteworthy that certain *CteR2R3-MYBs* demonstrated unique gene structures. This study revealed that the *CteR2R3-MYBs* may have the potential for functional diversification, due to the structural diversity of exons produced during the evolutionary process.

### 2.2. The Evolution and the Group Classification of R2R3-MYB Gene Family in C. teeta

To gain a deeper insight into the *R2R3-MYB* gene family in *C. teeta*, we constructed a phylogenetic tree using the neighbour-joining (NJ) method with 130 *R2R3-MYB* genes from *A. thaliana*, 88 *R2R3-MYB* genes from *C. teeta*, and 83 *R2R3-MYB* genes from *Coptis chinensis* Franch. (Figure 1). All *R2R3-MYB* genes were divided into 27 subgroups (designated as S1~S27). *CteR2R3-MYBs* were found in 24 of these subgroups, excluding S1, S3, and S22. Among them, S6 has the maximum number of family members with 7, while S2, S13, S15, and S27 each have 6 family members. Although the *R2R3-MYB* genes of *C. chinensis* have fewer members compared to the *CteR2R3-MYBs*, they were shown to be present in 25 subgroups. Overall, the evolutionary tree clustering patterns of the *R2R3-MYB* gene families remained highly similar between *C. teeta* and *C. chinensis* (Appendix A). In order to further understand the *R2R3-MYB* gene family of *C. teeta*, we also identified the members of the *R2R3-MYB* gene family of *Aquilegia coerulea* and *Kingdonia uniflora* using the same parameters. Meanwhile, we constructed a phylogenetic tree using protein sequences from 130 *R2R3-MYB* genes of *A. thaliana*, 88 *R2R3-MYB* genes of *C. teeta*, 83 *R2R3-MYB* genes of *C. chinensis*, 103 *R2R3-MYB* genes of *A. coerulea,* and 123 *R2R3-MYB* genes of *K. uniflora* (Appendix A). Based on the previous subgroup classification, the *R2R3-MYB* genes of *A. coerulea* and *K. uniflora* were finally determined to be distributed among 25 subgroups (Appendix A). S2, S3, and S11 displayed significant expansion in *A. coerulea* relative to the *CteR2R3-MYBs*. S2, S3, S4, S10, and S16 showed significant expansion in *K. uniflora*.

To clarify the evolutionary relationship between the *R2R3-MYB* gene family of *C. teeta* and other species, we performed collinearity analysis of the *C. teeta* genome with *A. thaliana*, *C. chinensis*, and *A. coerulea*, respectively (Figure 2). There were 23 collinear *R2R3-MYB* gene pairs in *C. teeta* and *A. thaliana*, 78 collinear *R2R3-MYB* gene pairs in *C. teeta* and *C. chinensis*, and 64 *R2R3-MYB* collinear gene pairs in *C. teeta* and *A. coerulea* (Appendix A). Based on the differences in the number of collinear *R2R3-MYB* gene pairs present between *C. teeta* and other species, we concluded that the kinship of the *R2R3-MYB* genes in *C. teeta* were *C. chinensis*, *A. coerulea*, and *A. thaliana*, in the order of strong to weak. This result was also consistent with the evolutionary relationships from the phylogenetic tree analysis.

To investigate the evolutionary conservation of co-linear *R2R3-MYB* gene pairs among four species (*A. thaliana*, *C. teeta*, *C. chinensis*, and *A. coerulea*), we analysed the *R2R3-MYB* gene pairs in *C. teeta* that display collinearity with other species, and found only 16 *R2R3-MYB* co-linear gene pairs co-existing in *C. chinensis*, *A. coerulea*, and *A. thaliana* (Figure 3, Appendix A). This suggested that the 16 *R2R3-MYB* genes were evolutionarily more conserved among the four species, with similar functional roles.

Duplication or loss of gene families frequently occurs in plants and results from whole-genome duplication (WGD) or whole-genome triple duplication (WGT) events [21,22]. To elucidate the evolution of the *R2R3-MYB* gene family in *C. teeta*, we first analysed gene duplication and loss in *C. teeta* and *A. thaliana* (Appendix A). It was found that the expansion of the *R2R3-MYB* gene family was exclusive to the ancestral lineages of *C. teeta* and *A. thaliana*. The number of *R2R3-MYB* genes that expanded in *A. thaliana* was nearly double the count of those in *C. teeta*. In addition, the number of *R2R3-MYB* genes lost in *C. teeta* was approximately twice that in *A. thaliana*. Furthermore, we expanded our analyses to include *Coptis chinensis*, belonging to the same genus as *C. teeta*; *A. coerulea*, a species within the same family as *C. teeta*, and *K. uniflora*, a species falling within the same order as *C. teeta*. The duplication and loss analysis was conducted using the five species (*A. thaliana*, *C. teeta*, *C. chinensis*, *A. coerulea*, and *K. uniflora*) (Appendix A). During the genealogy tracing of the common ancestor of the five species, 190 genes were duplicated, without any losses. Conversely, the number of lost *R2R3-MYB* genes was greater than the count of duplicated genes across the five species. These data lend support to the hypothesis that the *R2R3-MYB* genes in the majority of the plant species were lost subsequent to the whole-genome duplication (WGD) or whole-genome triplication (WGT) events, thereby enhancing functional diversity within *R2R3-MYB* gene family.

### 2.3. Chromosomal Location and Duplication Events of CteR2R3-MYBs

Among the 88 *CteR2R3-MYB* genes, 87 were shown to be unevenly distributed on 9 chromosomes, while *CteMYB88* was located on NO.699 unassembled contig (Figure 4). Chromosome 3 and chromosome 6 contained the largest number of *CteR2R3-MYBs* (12 genes, ~13.64%), followed by chromosome 9 (11 genes, ~12.50%), and chromosome 4 contained the smallest count *CteR2R3-MYBs* (5 genes, ~5.6%).

Gene duplication events in plants are known to facilitate the generation of novel genetic and functional diversity throughout growth and development, serving as a crucial mechanism propelling adaptive evolution and species diversification [23]. To further analyse the role of duplication events in the evolution of the *R2R3-MYB* genes in *C. teeta*, we identified 5 types of duplication events [24]. We obtained 177 duplicated gene pairs: 3 whole-genome duplicates (WGD duplicates, 1.69%), 3 tandem duplicates (TD, 1.69%), 163 dispersed duplicates (DSD, 92.09%), 3 proximal duplicates (PD, 1.69%), and 5 transposed duplicates (TRD, 2.82%) (Figure 5a, Appendix A).

To assess selective pressure and functional conservation of *CteR2R3-MYBs* during the evolutionary process, we compared the Ks, the Ka, and Ka/Ks values among groups of duplicated genes with 5 duplication modes (Figure 5b, Appendix A). Only 1 WGD gene pair exhibited higher Ka/Ks ratios and smaller Ks values, indicating rapid sequence divergence and stronger positive selection compared to genes originating through other duplication modes. The duplication modes of *CteR2R3-MYBs* revealed that the majority of gene pairs exhibited lower Ka/Ks ratios and smaller Ka values, indicating slower *R2R3-MYB* sequence divergence and stronger negative selection in *C. teeta*.

### 2.4. Analysis of cis-Acting Elements in CteR2R3-MYBs

To elucidate the transcriptional regulatory mechanism and explore the transcriptional pattern of the *CteR2R3-MYBs*, we predicted cis-acting elements upstream of the promoter regions of the genes. We predicted a total of 55 cis-elements in *CteR2R3-MYBs* (Appendix A, Appendix A), which play multiple roles in plant growth and development, such as hormone and light responses, metabolic regulation, and stress responses. Light-responsive and hormone-responsive elements were prevalent across the *CteR2R3-MYBs*, constituting 41% and 31%, respectively, of all response elements (Appendix A). The presence of hormone-responsive elements is crucial in floral development. Therefore, we conducted additional analyses to specify the distribution patterns of different hormone response elements within the *R2R3-MYB* genes. A total of 697 components were included in the hormone response, including (IAA) auxin responsiveness (71, 10.19%), (GA) gibberellin responsiveness (83, 11.91%), (SA) salicylic acid responsiveness (54, 7.75%), (ABA) abscisic acid responsiveness (253, 36.29%), and (MeJA) MeJA responsiveness (236, 33.86%). The majority of *CteR2R3-MYBs*, (86.36%), exhibited abscisic acid responsiveness (ABRE) in all promoter regions of *CteR2R3-MYBs*, while 64.77% demonstrated to MeJA responsiveness (CGTCA-motif and TGACG-motif), and 38.64% to GA responsiveness (GARE-motif, P-box, and TATC-box). Additionally, 86.36% of *CteR2R3-MYBs* contained ARE, a stress-responsive element. Certain *CteR2R3-MYBs*, such as *CteMYB28* and *CteMYB30*, were identified to harbour multiple hormone-responsive elements in their promoter regions, indicating a potentially heightened and more rapid response to specific hormones.

### 2.5. Prediction Regulatory Network of CteR2R3-MYBs

Utilizing the string database, the regulatory network of all CteR2R3-MYB proteins was predicted, with *A. thaliana* serving as a reference species. This network analysis enabled us to deduce the potential function of *R2R3-MYB* genes in the floral development (Figure 6). Interestingly, *CteMYB2*, *CteMYB18*, *CteMYB29*, *CteMYB2*, *CteMYB43*, *CteMYB69*, and *CteMYB74* exhibited a strong correlation to plant reproductive development. Their Arabidopsis counterparts predominantly participated in anther dehiscence, pollen maturation, and female gametophyte formation. These findings indicated that the *R2R3-MYB* genes may be involved in regulating the sexual reproduction in *C. teeta*.

### 2.6. Dichogamous and Herkogamous Expression Pattern of CteR2R3-MYB Genes

Previous research has demonstrated the widespread involvement of *R2R3-MYBs* in plant sexual reproduction [18]. However, there is a scarcity of studies investigating the possible regulatory function of the *R2R3-MYB* gene family in dichogamy and herkogamy. Thus, we utilized the blastp program to identify homologous *CteR2R3-MYBs*, using genes associated with sexual reproduction in *A. thaliana* (Appendix A). We identified 19 *CteR2R3-MYBs* as candidate genes, which were homologous to those found in *A. thaliana*.

The string database can help understand patterns of protein interactions and thereby helps predict protein function and reveals potential pathways in biological processes. Seven *R2R3-MYB* genes were identified from the *CteR2R3-MYB* genes that were potentially associated with the development of sexual reproduction in plants through the Protein-Protein Interaction Network (Figure 6). High gene expression ensures an ample supply of gene products for executing specific biological functions. We analysed *CteR2R3-MYB* genes with high expression, excluding the genes predicted by the string database and blastp homology comparisons. These findings may indicate that the identified *MYB* genes could be involved in the development of diverse flower types in *C. teeta*, based on in predictions and qPCR validations, although further functional analyses are necessary to confirm their roles.

To explore the molecular mechanism(s) underlying dichogamy and herkogamy, we examined two types of floral tissues from *C. teeta*: type I (Maternal type) is protogynous with exserted stigmas, and type II (Paternal type) is protandrous with concealed stigmas. Then, we selected three time periods for analysis: P0; when the floral organs were fully enclosed in bracts, a significant difference in pistil length was observed between the two types, with the pistils of the Maternal type being longer than the Paternal type. P1; the bracts partially exposed the floral organs, and the pistil continued to elongate within the Maternal type. P2; the bracts were fully exposed, petals began to unfold gradually, and the pistil continued to elongate in the Maternal type, with minimal change in stamen length. In contrast, the stamen length slightly increased in the Paternal type, and the pistil length was significantly shorter than the stamen length. Gene expression can vary considerably between organs and physiological states, reflecting the level of gene activity in the organism and thereby showcasing their function. Comparing the expression levels of *CteR2R3-MYB* genes during different stages helps to identify *R2R3-MYB* genes associated with the development of the two floral phenotypes observed in *C. teeta* (dichogamous and herkogamous).

To elucidate the expression patterns of *CteR2R3-MYB* genes during floral development, we first analysed the expression pattern of 19 genes in both floral phenotypes during three selected time periods using RNA-Seq data (Figure 7a). *CteMYB29* was highly expressed in P_P2, while *CteMYB86*, *CteMYB48*, *CteMYB46*, *CteMYB15*, and *CteMYB74* were highly expressed in M_P2. We then examined the expression of *CteR2R3-MYB* genes that were predicted to participate in the regulatory network (Figure 7b). *CteMYB69* exhibited high levels of expression in M_P0, while *CteMYB2* showed high expression in P_P0. Furthermore, among the highly expressed genes, we identified 4 genes with high expression levels in M_P0 and 3 genes highly expressed in P_P0. We identified two sets of candidate genes from the mentioned genes: Group 1: 7 *CteR2R3-MYB* genes homologous to those in *A. thaliana* with verified functions (*CteMYB2*, *CteMYB15*, *CteMYB25*, *CteMYB29*, *CteMYB46*, *CteMYB69*, and *CteMYB74*); and Group 2: 10 *CteR2R3-MYB* genes with unknown functions that exhibited significant differences between periods and types (*CteMYB4*, *CteMYB7*, *CteMYB16*, *CteMYB24*, *CteMYB33*, *CteMYB45*, *CteMYB54*, *CteMYB70*, *CteMYB71*, and *CteMYB72*).

### 2.7. Gene Expression Validation (qRT-PCR)

To validate the results of the RNA-Seq, we quantified the expression levels of *CteR2R3-MYB* genes in different floral phenotypes and time periods using qRT-PCR (Figure 8). The results showed that the expression levels of RNA-Seq and qRT-PCR were consistent.

## 3. Discussion

*Coptis teeta* Wall., a perennial herb of the genus *Coptis* in the *Ranunculaceae* family of the order *Ranales*, exhibits simultaneous asexual and sexual reproduction. As a monocious plant, it employs both dichogamous and herkogamous reproductive strategies. Two floral phenotypes are widespread within wild populations of this species: Type I (Maternal type) is protogynous with exserted stigmas, and type II (Paternal type) is protandrous with concealed stigmas. The traits of dichogamy and herkogamy play an important role in plant sex determination and flower development, which have important implications for plant reproduction and seed production. Members of the *R2R3-MYB* gene family may be involved in the formation of these traits, and the molecular mechanism can be initially explored by analysis of the members of *R2R3-MYB* gene family in *C. teeta*, which can help to understand the reproductive biology of this economically important species.

### 3.1. Evolutionary Analysis of the CteR2R3-MYB Gene Family

This study identified 88 *R2R3-MYB* family genes from the genome of *C. teeta* and elucidated their fundamental features, conserved motifs, and phylogenetic relationships with *C. chinensis* and *A. thaliana*. In addition, we analysed *R2R3-MYB* gene families of four species in the order *Ranales* which had been sequenced in recent years. A systematic comparative analysis was conducted from an evolutionary standpoint, encompassing gene duplications and losses, as well as evolutionary and phylogenetic relationships.

The number of members of the *R2R3-MYB* gene family in *C. teeta* is close to that identified in *C. chinensis* using the same methods, but differs markedly from *A. thaliana*. The number of *R2R3-MYB* gene family members showed a gradual increase among *C. chinensis* (83), *C. teeta* (88), *A. coerulea* (103), *K. uniflora* (123), and *A. thaliana* (133). The substantial expansion of the *R2R3-MYB* gene family is believed to stem from genome-wide duplication, segmental duplication, and tandem duplication [25]. The fewer members of the *R2R3-MYB* gene family of *Coptis* suggests a relatively minor number of duplication events during the evolutionary process. The dispersed duplication (DSD) events play a crucial role in the expansion of *MYB* gene family among the five species studied. Collinearity analysis highlighted a remarkable consistency between the changed numbers of *MYB* collinearity gene pairs and the established phylogenetic relationships among the species.

The Ka/Ks ratio serves as a representation of selective pressures during biological evolution [26]. Analysis of Ka/Ks ratios for all the identified homologous pairs, encompassing 93 *MYB* gene pairs, indicated that the majority of *CteR2R3-MYBs* have undergone purifying selection, implying their involvement in highly conserved evolution. Furthermore, positive selection was observed in only one *CteMYB23*, indicating its functional novelty.

Our analyses showed that more *R2R3-MYB* family genes were detected in *A. thaliana* than in the family *Ranunculaceae*. Additionally, expansions were observed in the common ancestor nodes of these species, along with species-specific expansions relative to their close relatives. The *MYB* gene losses significantly surpassed the gene duplications in the identified species. This suggested that the gene losses of the members of the *MYB* gene family after duplication event might lead to the functional diversity of the remaining *MYB* genes in these species.

### 3.2. CteR2R3-MYBs Might Play Crucial Roles in the Hormone Regulation of Floral Development

Phytohormones such as GA, IAA, and JA [16,27,28,29] play a crucial role in the development of floral sex organs in plants, serving as vital signalling molecules that regulate plant sexual development. Analysis of gene promoter cis-acting elements in the *CteR2R3-MYBs* revealed an enrichment of various hormone-responsive cis-acting elements within the promoter region. The type and quantity of these elements could impact gene expression levels and biological functions. In all groups of the candidate genes identified (Figure 5; Appendix A), a variety of hormone-responsive cis-acting elements were detected. Particularly noteworthy was the presence of abscisic acid responsiveness in each gene’s cis-acting elements. In addition, there were more cis-acting elements for MeJA responsiveness (40.79%) in the candidate genes. Moreover, there was a higher abundance of gibberellin responsiveness (14.26%) and salicylic acid responsiveness (14.26%) than for other cis-acting elements. It has been shown that normal accumulation of salicylic acid promotes the degeneration of stamens in female flowers, resulting in the formation of structurally and functionally normal female flowers [30]. During the later stages of stamen development, it has been demonstrated that GA stimulates the production of JA [31]. Consequently, JA triggers the activation of three *MYB* genes that are essential for the growth of the filament and for the maturation of the anthers [32]. These results may indicate the potential involvement of abscisic acid, gibberellin, salicylic acid, and methyl jasmonate in the sexual reproductive development of *C. teeta*. *R2R3-MYB* genes play crucial roles in floral development

*MYB* transcription factors are pivotal in the sex-biased expression of flowers in *C. teeta*, as evidenced by their high expression levels and their involvement in the regulatory network alongside NAC family members. In order to screen for *CteR2R3-MYB* genes with the function of regulating sexual reproduction in *C. teeta*, we analysed the expression levels of *CteR2R3-MYB* genes in two types of *C. teeta* flowers during different periods. A total of 17 highly expressed candidate genes were identified, including seven *CteR2R3-MYB* genes with predicted functions and ten genes with unknown functions. To determine the regulatory roles of *R2R3-MYB* members in sexual reproduction in *C. teeta*, we identified *CteR2R3-MYB* genes with regulatory roles by blastp and string database analyses (Figure 6, Appendix A). Among these genes was the homologue of *CteMYB2* in *Arabidopsis* which was identified as *AtMYB26* (*AT3G13890*). *AtMYB26* protein exhibits specific localization within the wall nucleus of the anther chamber, where it directly regulates the expression of two NAC genes, *NST1* and *NST2* [33]. These genes are pivotal for the induction of anther secondary thickening in *A. thaliana* [34]. *CteMYB2* exhibited high levels of expression during both the P0 and P2 time periods in the paternal-type flowers. This result aligns with previous studies, indicating that *CteMYB2* (orthologous to AtMYB26) might be linked to the development of plant male reproductive organs.

In conclusion, we propose that these seven *CteR2R3-MYBs* with predicted functions might be most likely implicated in the formation of diverse flower types in *C. teeta* (Figure 9). In addition to candidate genes with established functions, we also selected 10 genes with unknown functions which displayed differential expression during the selected time periods and in the two floral phenotypes (Figure 9). These genes merit further investigation to gain a deeper understanding of the formation of dichogamy and herkogamy, including the possibility of gene neofunctionalization in the *R2R3-MYB* gene family.

## 4. Materials and Methods

### 4.1. Identification and Sequence Analysis of CteR2R3-MYB Genes in C. teeta

Protein sequences of all reported *AtR2R3-MYBs* in *A. thaliana* were retrieved from the TAIR database (https://www.arabidopsis.org/, accessed on 20 February 2023). The *C. teeta* genomic data were obtained from our own research (data not published). Candidate *CteR2R3-MYBs* were initially identified in the *C. teeta* genome using a BLASTP search with *AtR2R3-MYBs* as seed sequences under e < 10^−5^. The Hidden Markov Model (HMM) file of the MYB DNA binding domain (PF00249) was downloaded from the Pfam database (http://pfam.xfam.org/, accessed on 20 February 2023) to identify *CteR2R3-MYBs*. Finally, candidate *CteR2R3-MYB* genes were identified by intersecting genes identified using two methods. All candidate *CteR2R3-MYBs* were predicted using SMART (http://smart.embl-heidelberg.de/, accessed on 22 February 2023) and NCBI-CDD online software (https://www.ncbi.nlm.nih.gov/Structure/, accessed on 22 February 2023) to filter out incompletely conserved domains. Theoretical isoelectric point, molecular weight, and other basic *CteR2R3-MYB* gene information was predicted using ExPaSy (https://web.expasy.org/protparam/, accessed on 23 February 2024). CELLO (http://www.csbio.sjtu.edu.cn/bioinf/plant/, accessed on 23 February 2024) was used to predict the subcellular location of *CteR2R3-MYBs* [38].

### 4.2. Analysis of CteR2R3-MYBs Conserved Motif and Gene Structure

Conserved protein motifs were predicted using the MEME (https://meme-suite.org/meme/, accessed on 24 February 2024) with a maximum of 10 motifs, while other parameters were set to default. The chromosomal locations and exon-intron structures of the *CteR2R3-MYB* genes were obtained from the genome annotation file. Visualization was performed using CFVisual (v 2.1.5) [39].

### 4.3. Phylogenetic Tree Construction and Collinearity Analysis

Protein sequences of *CteR2R3-MYB* gene homologs from the following five species were used in the phylogenetic analysis: *Arabidopsis thaliana* (*A. thaliana*, Ath), *Coptis teeta* (*C. teeta*, Cte), *Coptis chinensis* (*C. chinensis*, Cch) [40], *Aquilegia coerulea* (*A. coerulea*, Aco) [41], and *Kingdonia uniflora* (*K. uniflora*, Kun) [42]. Species with close phylogenetic relationships with *C. teeta* and species for which a large amount of basic data on MYB gene research already existed (*Arabidopsis thaliana*) were selected for further comparative analyses. *K. uniflora*, which is in the same genus (*Coptis chinensis* Franch.), family (*A. coerulea*), and dioecious (*Ranunculaceae*) as *C. teeta*, was also selected to ensure the biological relevance of the comparison. These species included other herbs and representative plants of the same family. Model plants with detailed data in *R2R3-MYB* gene studies, such as *Arabidopsis thaliana*, were also selected. These species are widely used in studies of gene function and expression patterns and provide a reliable reference for our comparative analysis. All species selected had high-quality genomic data to ensure the accuracy and reproducibility of the comparative analysis.

All the R2R3-MYB sequences were aligned using Muscle5 [43]. The phylogenetic tree was constructed using the neighbour-joining (NJ) method, using 1000 bootstrap replicates. Using MEGA11, the phylogenetic tree was examined and modified [44]. The syntenic relationships between the *CteR2R3-MYB* genes and *R2R3-MYB* genes from *A. thaliana*, *C. chinensis*, and *A. coerulea* were determined using TBtools (v 2.056) [45]. Duplication and loss of *R2R3-MYB* genes was identified using TBtools (v 2.056) and Notung software (v 2.8) [46]. All the phylogenetic tree results were visualized by the iTOL online program (https://itol.embl.de/, accessed on 1 April 2023) [47].

### 4.4. Chromosomal Location and Duplication Mode Analysis of CteR2R3-MYB Genes

The genomic coordinates of *CteR2R3-MYBs* loci were retrieved from the genome annotation files. The distribution of *CteR2R3-MYBs* on the chromosomes was mapped and the gene density was calculated and outputted by TBtools (v 2.056) [45]. Moreover, we utilized DupGen_finder with unique parameters to characterize various patterns of gene duplication, encompassing whole-genome duplication (WGD), tandem duplication (TD), proximal duplication (PD, less than 10 genes on the same chromosome), transposed duplication (TRD), and dispersed duplication (DSD, other than WGD, TD, PD, and TRD) [22]. Additionally, the Ka (non-synonymous substitution rate), Ks (synonymous substitution rate), and their ratios (Ka/Ks) were calculated utilizing the Simple Ka/Ks Calculator (NG) of TBtools according to the coding sequences (CDS) and protein sequences [45].

### 4.5. Prediction of Promoter cis-Elements and the Regulatory Network of CteR2R3-MYBs

PlantCare (https://bioinformatics.psb.ugent.be/webtools/plantcare/html/, accessed on 19 February 2024) was used for cis-element prediction based on the promoter sequences of *CteR2R3-MYBs* (located 2kb upstream of the promoter codon) [48]. *A. thaliana* was used as the reference sequence for string (https://string-db.org/, accessed on 23 February 2024) prediction network interaction maps [49]. Network diagram was visualized using cytoscape (v 3.9.1) [50].

### 4.6. RNA-seq Analysis

The sequencing plants were from the same population, planted in 2013 at Coptis teeta Wall. Cultivation Base, Zhiziluo Village, Nujiang, Yunnan Province, under the management of Foveolin Biotechnology Co., to ensure consistency of genetic background. Different types of whole flowers were used as samples for RNA extraction. Each sample was extracted from 10 flowers and each experimental replicate contained flowers from 10 different plants. From previously published studies, we obtained transcriptomic data for three time periods and two floral phenotypes (M and P) in *C. teeta*.

These transcriptome datasets were downloaded from NCBI and associated with project numbers PRJNA1127536 and PRJNA973818. The expression levels of the *CteR2R3-MYBs* were analysed using TPM values (Appendix A), and we generated an expression heatmap using TBtools (v 2.056).

### 4.7. RNA Isolation and qRT-PCR

The total RNA of samples was extracted from two floral phenotypes types (Maternal type and Paternal type) at time periods P0, P1, and P2 using a kit, according to the manufacturer’s instructions (Magen, Guangzhou, China). The concentration of the extracted total RNA was measured using the NanoDrop technique (Thermo Fisher Scientific, Waltham, MA, USA), the equivalent amount RNA was reverse transcribed into cDNA using a reverse transcription kit (TAKARA, Beijing, China). Five genes were chosen to verify the RNA-seq data. Quantitative real-time PCR (qRT-PCR) was performed using an Applied Biosystems QuantStudio 5 system (Thermo Fisher Scientific, Waltham, MA, USA) with the ChamQ Universal SYBR qPCR Master Mix (Vazyme, Nanjing, China). The PCR reaction was performed as follows: pre-denaturation 95 °C for 30 s, denaturation at 95 °C for 30 s, and annealing at 58 °C for 30 s, and this reaction was repeated for 40 cycles. Subsequently, an extra procedure was conducted as follows: denaturation at 95 °C for 15 s, annealing at 60 °C for 1 min, and extension at 72 °C for 15 s. The primers are listed in Appendix A. The relative expression of each gene was analysed using the 2^−ΔΔCT^ method with *C. teeta EF1α* gene as the reference gene [51]. Standard errors were calculated based on three biological replicates.

## 5. Conclusions

A total of 88 *CteR2R3-MYB* genes were identified in this study, and the physicochemical characteristics, subcellular localisation, gene structure, conserved motifs, chromosomal location, and phylogeny of replication events of these genes were analysed. We also performed a preliminary analysis of the evolution of the *R2R3-MYB* gene family in the order *Ranales*. In addition, we analysed the promoter cis-acting elements of the *CteR2R3-MYB* genes to elucidate the potential roles of hormone-responsiveness in the development of *C. teeta* flowers. By analysing the tissue-specific expression pattern of *CteR2R3-MYB* genes, two groups of candidate genes were identified. Taken together, the results of this study can be used to explore the involvement of *CteR2R3-MYB* genes in the development of dichogamy and herkogamy in *C. teeta*, these data also provide a useful resource for studying the biological functions and evolutionary history of the *R2R3-MYB* gene family in the order *Ranales*.

In summary, our study provides a comprehensive analysis of the *R2R3-MYB* gene family in *C. teeta*, highlighting their potential roles in various biological processes. Notably, we identified seven *CteR2R3-MYBs* that are most likely implicated in the formation of diverse flower types in *C. teeta*. These findings suggest specific regulatory roles of these *MYBs* in floral development, contributing to our understanding of the genetic mechanisms underlying flower diversity in this species.

## Figures and Tables

**Figure 1 ijms-25-08902-f001:**
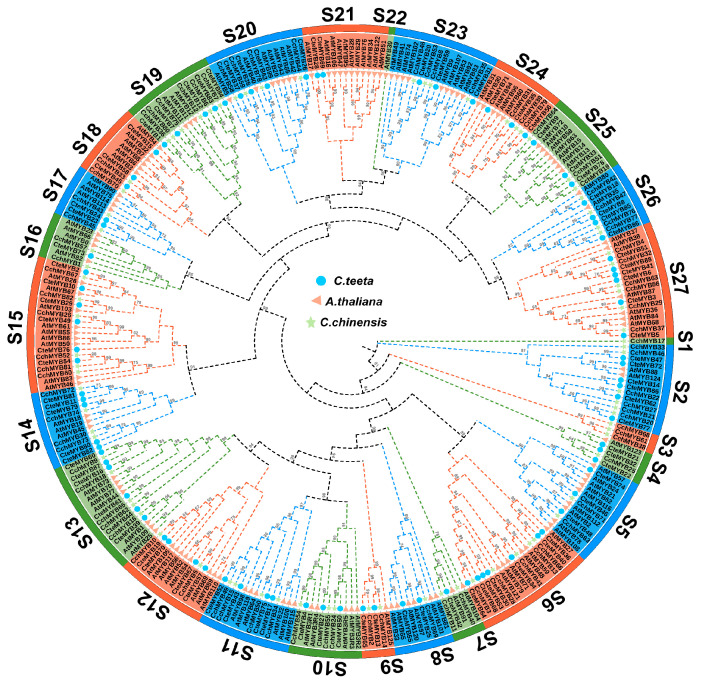
Phylogenetic relationships of *R2R3-MYBs*. *A. thaliana*, *C. teeta*, and *C. chinensis R2R3-MYBs* were used for phylogenetic tree construction using the neighbour-joining method. Blue circles denote the *R2R3-MYBs* of *C. teeta*, orange triangles represent the *R2R3-MYBs* of *A. thaliana*, and green pentagrams symbolize the *R2R3-MYBs* of *C. chinensis*.

**Figure 2 ijms-25-08902-f002:**
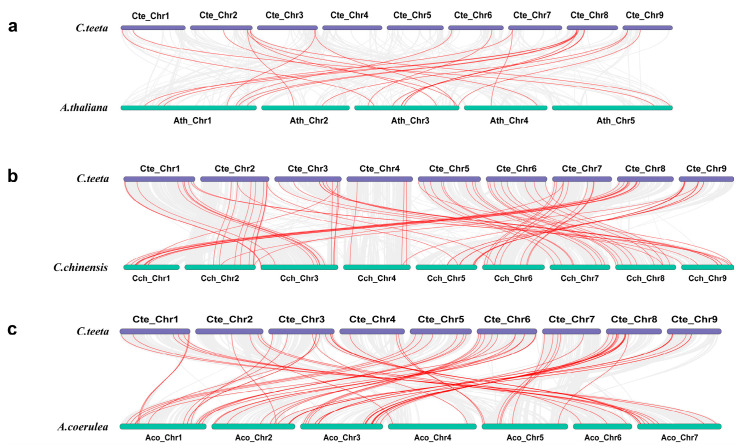
Syntenic relationship within the R2R3-MYB family of *A. thaliana*, *C. teeta*, *C. chinensis*, and *A. coerulea*. Collinear blocks are depicted by gray lines in the genomes of *C. teeta* and other plants, while collinear *R2R3-MYB* gene pairs are highlighted by red lines. (**a**) Analysis of inter-species co-linear between *Arabidopsis thaliana* and *Coptis teeta* Wall. (**b**) Analysis of inter-species co-linear between *Coptis chinensis* and *Coptis teeta* Wall. (**c**) Analysis of inter-species co-linear between *Aquilegia coerulea* and *Coptis teeta* Wall.

**Figure 3 ijms-25-08902-f003:**
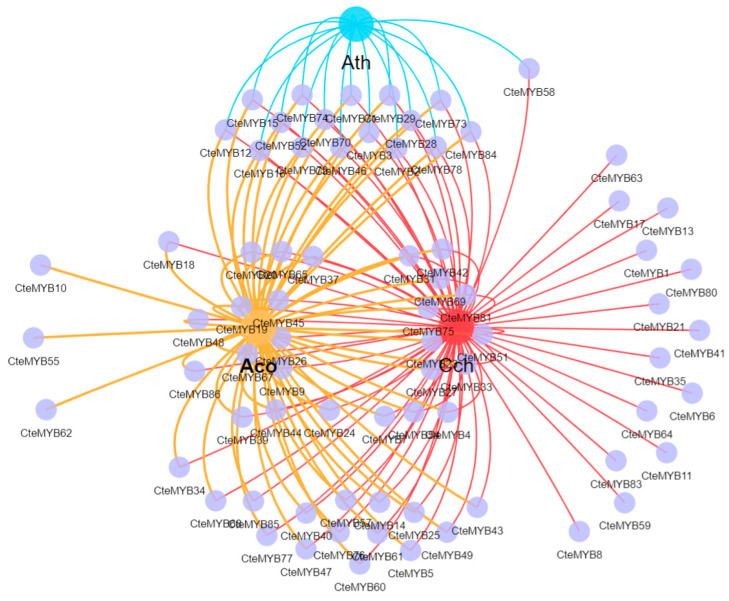
The collinear relationships of *CteR2R3-MYBs* with each of the three species. Points correspond to a unique gene participating in synteny. Lines connecting circles denote the presence of syntenic gene pairs within the respective species. Lines enclosed within circles indicate the occurrence of syntenic genes across species. A single connecting line denotes that the gene is syntenic exclusively within that species. Ath: *Arabidopsis thaliana*, Aco: *Aquilegia coerulea*, Cch: *Coptis chinensis*. Collinear gene pairs were identified using TBtools (v 2.056) and visualized via CNSKnowall online (https://www.cnsknowall.com/, accessed on 28 February 2024).

**Figure 4 ijms-25-08902-f004:**
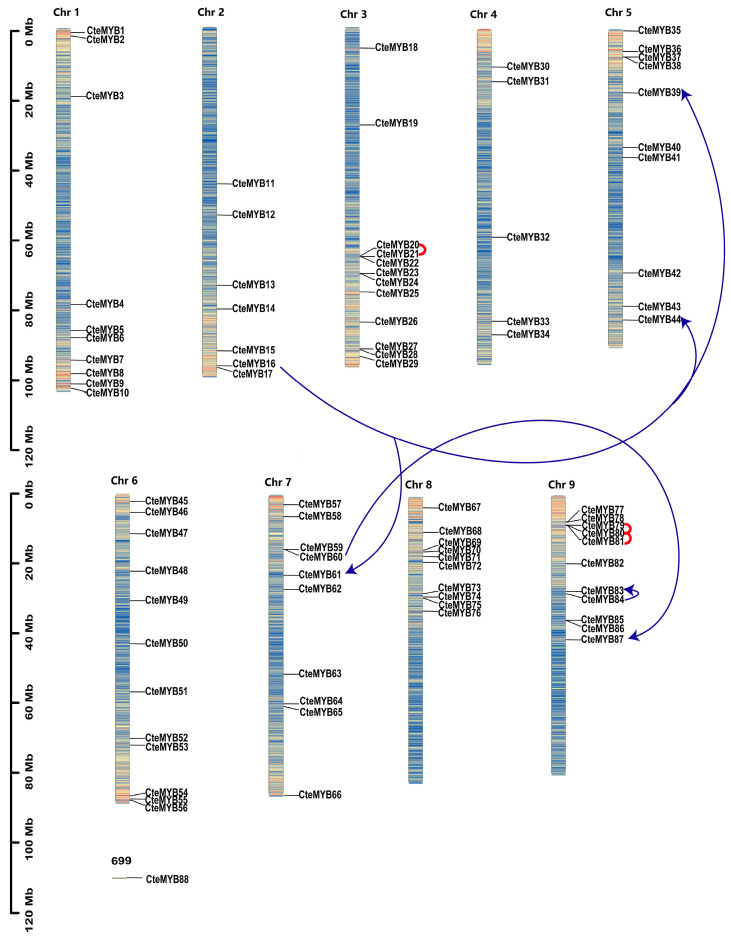
Chromosomal distribution of *R2R3-MYB* genes on 9 chromosomes of *C. teeta*. Distribution of *R2R3-MYB* genes on 9 chromosomes in *C. teeta*. Tandemly and proximally duplicated genes are indicated by red lines, respectively. Transposed duplicated genes are indicated by blue lines, arrow points denote transposed genes.

**Figure 5 ijms-25-08902-f005:**
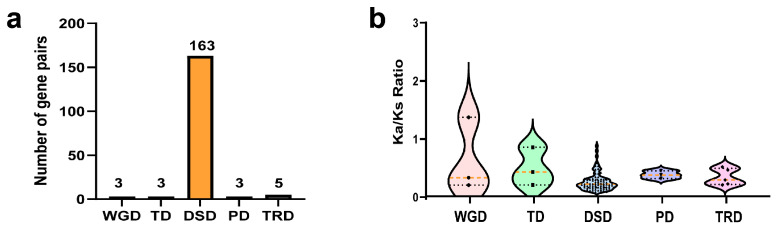
(**a**) Distribution of gene pairs with different duplication models of *CteMYBs*. (**b**) The Ka/Ks ratio distributions of gene pairs derived from five modes of duplication. The centre line is the median; the lower and upper dotted line correspond to the first and third quartiles (25th and 75th percentiles). WGD, whole-genome duplication; TD, tandem duplication; DSD, dispersed duplication; PD, proximal duplication; TRD, transposed duplication.

**Figure 6 ijms-25-08902-f006:**
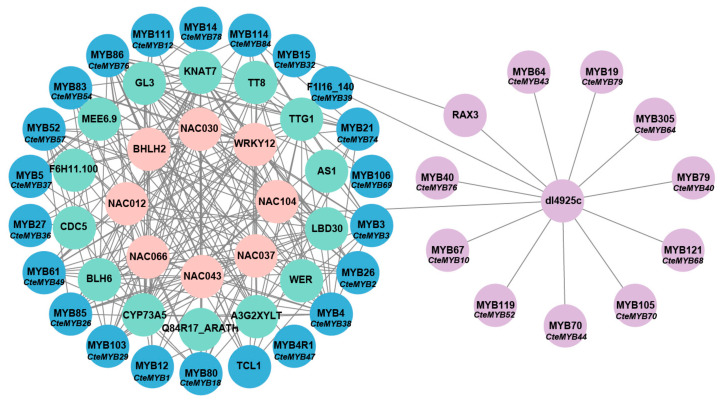
Protein regulatory network of *CteR2R3-MYB* genes by string database. The selected *CteMYBs* are located on the outermost side, with possible detection of proteins inside. Some of the blue circles and purple circles represent the *CteMYBs* used in this study, and the grey lines represent possible regulatory relationships. Where the gene names above each circle represent Arabidopsis genes whose functions can be queried in detail in the string database, and the ones below correspond to *CteMYBs*.

**Figure 7 ijms-25-08902-f007:**
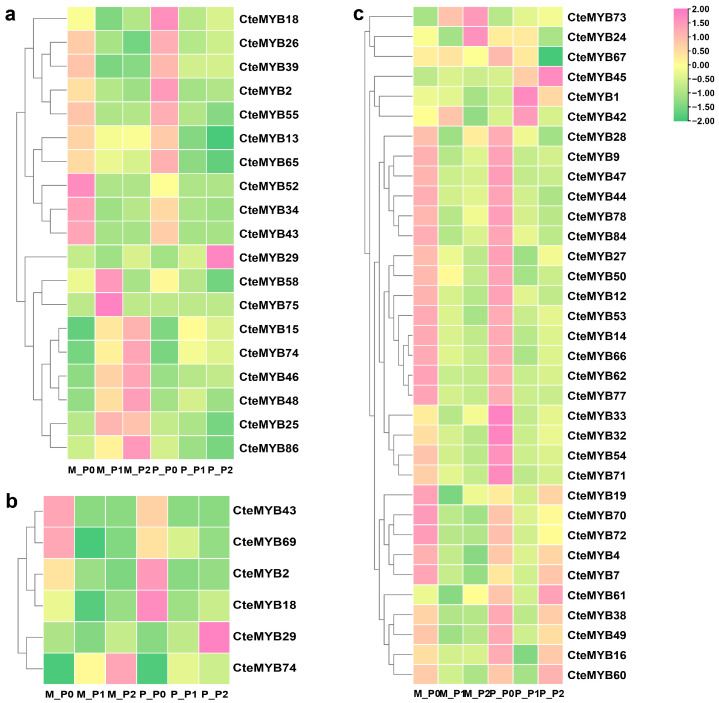
Expression patterns of selected *CteR2R3-MYBs*. (**a**) potential *CteR2R3-MYBs* identified by blastp. (**b**) *CteR2R3-MYBs* shown to potentially be involved in the regulatory network by string database analyses. (**c**) *CteR2R3-MYBs* with high expression at different time periods. M denotes the Maternal flower type; P denotes the Paternal flower type. Transcriptome expression data are normalized. Each point represents the mean of three independent biological replicates. M: Maternal type; P: Paternal type; P0/P1/P2: different developmental periods.

**Figure 8 ijms-25-08902-f008:**
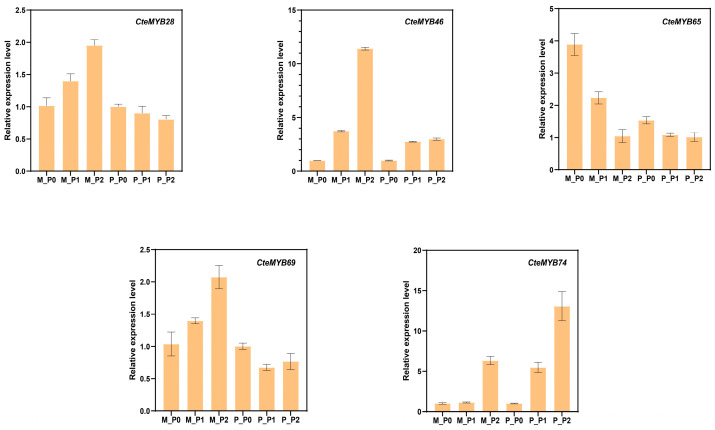
Validation of selected genes by qRT-PCR. M: Maternal type; P: Paternal type; P0/P1/P2: different developmental periods.

**Figure 9 ijms-25-08902-f009:**
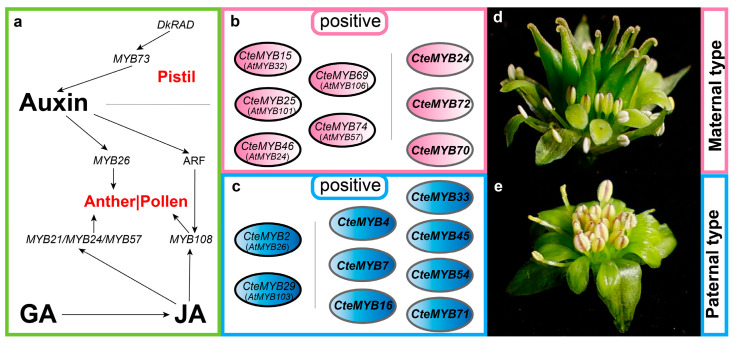
Illustration of the hypothetical roles of *R2R3-MYBs* in the regulation of dichogamy and herkogamy in *C. teeta*. (**a**) The role of hormones in plant pistil and stamen development [18,31,34,35,36,37]. GA: gibberellic acid; JA: jasmonic acid; *DkRAD*: a small-MYB RADIALIS-like gene in persimmons, overexpression of *DkRAD* in model plant resulted in hypergrowth of the gynoecium; *MYB73*: a *MYB* gene in persimmons; *MYB26*: a MYB gene in *Arabidopsis*, key to spatial specificity in anther secondary thickening formation; *MYB108*: a *MYB* gene in *Arabidopsis*, silencing of the transcription factor *MYB108* delays anther dehiscence and reduces pollen viability; *MYB21/MYB24/MYB57*: MYB gene in *Arabidopsis*, critical for *Arabidopsis* stamen development. (**b**) The *R2R3-MYB* genes that might have a positive role in Maternal-type flower development in *C. teeta*. Black bordered ellipses represent putative known functions of the *CteR2R3-MYBs*; ellipses with grey borders represent unknown functions of the *CteR2R3-MYBs*. (**c**) *R2R3-MYB* genes that might have a positive role in Paternal-type flower development in *C. teeta*. Black bordered ellipses represent putative known functions of the *CteR2R3-MYBs*; ellipses with grey borders represent unknown functions of the *CteR2R3-MYBs*. (**d**) The floral phenotype of Maternal-type flowers in *C. teeta*. (**e**) The floral phenotype of Paternal-type flowers in *C. teeta*.

## Data Availability

All data are contained within the article and Appendix A.

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
