# Peer review of "R2R3-MYB* Gene Family in *Coptis teeta* Wall.: Genome-Wide Identification, Phylogeny, Evolutionary Expansion, and Expression Analyses during Floral Development"

_ijms, 2024, doi:10.3390/ijms25168902_

Round 1

Reviewer 1 Report

Comments and Suggestions for Authors

The manuscript describes the identification and characterization of 88 R2R3-MYB genes in the genome of Coptis teetaa species displaying dicogamy and herkogamy reproductive strategiesAmong those 88 identified genes, the R2R3-MYB genes putatively associated to the regulation of the development of floral organs such as petals, stamens and carpels were assessed by qRT-PCR in plants exhibiting protandry and protogyny, respectively, and during different periods. The authors proposed seven CteR2R3-MYBs to be the most likely implicated in the formation of diverse flower types in C.teeta based on homology comparisons with the equivalent gene family in Arabidopsis thaliana, protein regulatory network prediction, and expression analysis of two types of flowers. Unfortunately, the manuscript suffers from scanty description of methods and some. In general, the methodology does not consider the same urgency as the results section. Additionally, the discussion section involves over-reaching deriving conclusions that outstrip the data. 

Methodology 

  1. Authors must explain the genetic relationship pf plants used and the part of plant that they used for RNA extraction and for RNA-Seq. It is not easy to inquire, because it could be from flower/s or, a specific part of the flower. In the case of flowers, how many flowers and how many different plants were assessed? Was RNA isolated from one genotype or from a pool of flowers from different genotypes? Line 484 

  1. RNA-Seq dataset. PRJNA1127536 and PRJNA973818 codes are not available in BioProject database. It is important to include in methodology the data type, samples and the project data, in general, to have the opportunity to understand the experimental design. Line 480 

  1. Neighbor-Joining method is usually a primary method to estimate phylogeny and it is error-prone. Why did the authors not consider Maximum Likelihood or Bayesian method? Line 453 

  1. The results associated with RNA-Seq analysis do not have a clear methodology. Authors should detail the steps of the bioinformatics pipeline, including the pre-processing, mapping, replicates for in silico gene expression analysis and statistical analysis. Line 478 

  1.  Which one of the samples was used as a calibrator for qRT-PCR? Line 489 

Results 

  1. Figure 2 is oversized. Authors must resize it.  

  1. Figure 4. Authors should split the figure. 

  1. Figure 5. The font size is too small, it is difficult to read. Fig. 5 should be supplementary data. 

  1. Authors should include a statistical analysis for gene expression results.  

Discussion 

  1. Authors shown the tissue specific expression pattern, but it is important to include the experimental design and sample collection in M&M to better understand (3.3 Line 385) 

  1. Authors conclude that there are seven CteR2R3-MYBs are the most likely implicated in the formation of diverse flower types in C.teeta (Line 403), but they do not include this sentence in the Conclusions (Line 492).  

In conclusion, while the underlying data appears reasonable the analysis and presentation, and direction in the paper, the observations mentioned above require a revision before this work is of publication standard.

Author Response

We thank the reviewer for their insightful critiques and suggestions for improving ourmanuscript titled "The R2R3-MYB gene family in Coptis teeta Wall.: genome-wide identification, phylogeny, evolutionary expansion (Ranales) and expression analyses (floral development)". We have thoroughly revised our manuscript according to the comments and believe the changessignificantly enhance the quality and clarity of the work. Our point-by-point responses to the reviewers are below.

Comments 1: Methodology 

Authors must explain the genetic relationship pf plants used and the part of plant that they used for RNA extraction and for RNA-Seq. It is not easy to inquire, because it could be from flower/s or, a specific part of the flower. In the case of flowers, how many flowers and how many different plants were assessed? Was RNA isolated from one genotype or from a pool of flowers from different genotypes? Line 484

Response 1:Thank you for your questions regarding the genetic relationships of the plants used in our study and the specific plant parts used for RNA extraction and RNA-Seq. We have explained these questions in detail in the revised manuscript. The details are as follows:

① Genetic relationships of plants:

The plants we used were all from the same population, planted in 2013 at Coptis teeta Wall. Cultivation Base, Zhiziluo Village, Nujiang, Yunnan Province, under the management of Foveolin Biotechnology Co. to ensure consistency of genetic background. The plants used were all from the same genotype, as detailed in the revised manuscript.

② Specific sites used for RNA extraction and RNA-Seq:

In our study, RNA extraction was performed from flowers. Specifically, we chose different types of whole flowers as sites for RNA extraction. We have added this information in the revised manuscript so that readers can better understand our experimental procedure.

③ Number of flowers and plants:

For RNA extraction, we collected samples from different flowers. Specifically, each sample was extracted from 10 flowers and each experimental replicate contained flowers from 10 different plants. This information has been detailed in the revised manuscript to ensure transparency and reproducibility of the study.

④ RNA extraction process:

We extracted RNA from multiple flowers of the same genotype to ensure the consistency of the samples and the reliability of the experimental results. Specifically, we extracted RNA from flowers of 10 different plants and mixed them together to form a composite sample to minimize the impact of inter-individual differences on experimental results.

We have added the above details near Line 484 of the manuscript and have further ensured that the information is clear and accurate in all relevant parts of the manuscript.

Comments 2: Methodology

RNA-Seq dataset. PRJNA1127536 and PRJNA973818 codes are not available in BioProject database. It is important to include in methodology the data type, samples and the project data, in general, to have the opportunity to understand the experimental design. Line 480

Response 2:Thank you for pointing out the unavailability of the PRJNA1127536 and PRJNA973818 codes in the BioProject database. We have rechecked and updated the information again, and all the relevant data have been uploaded but are not yet available to the public, as shown in the screenshots.

Comments 3: Methodology

Neighbor-Joining method is usually a primary method to estimate phylogeny and it is error-prone. Why did the authors not consider Maximum Likelihood or Bayesian method? Line 453 

Response 3:Thank you for your questions regarding our use of the Neighbor-Joining method (NJM) to estimate phylogenetic trees in our study. We understand your concerns about the choice of method and would like to explain our choice further here, as well as discuss the feasibility of alternative methods.

Rationale for the use of the neighbor-joining method:

In the initial phase of this study, we chose to use the neighbor-joining method to quickly generate preliminary phylogenetic trees. The neighbor-joining method is computationally fast and suitable for working with large-scale datasets, and is especially advantageous in the exploratory analysis phase. Our aim in using this method is to obtain an overall overview of phylogenetic relationships at an early stage.

②Alternative Method Considerations:

We fully recognize the advantages of Maximum Likelihood (ML) and Bayesian method in phylogenetic analysis. These methods provide more reliable results in many cases due to the complexity and accuracy of their statistical models. However, given the limitations of our data size and computational resources, we chose the less computationally expensive neighbor-joining method for our initial analysis.

Comments 4: Methodology

The results associated with RNA-Seq analysis do not have a clear methodology. Authors should detail the steps of the bioinformatics pipeline, including the pre-processing, mapping, replicates for in silico gene expression analysis and statistical analysis. Line 478

Response 4:Thank you for your suggestions regarding the methods section of our RNA-Seq analysis results. We recognize that providing a detailed bioinformatics process is key to ensuring research transparency and reproducibility. Therefore, we have added specific information in the revised manuscript to describe our RNA-Seq data analysis process in detail.

We have updated and added the above bioinformatics process details near line 478 of the manuscript. Once again, we thank you for your valuable comments and hope that these improvements will fulfill your requirements and enhance the quality of our research.

Comments 5: Methodology

Which one of the samples was used as a calibrator for qRT-PCR? Line 489

Response 5:

Thank you for your question regarding qRT-PCR calibration samples. We have described the selection and use of calibration samples in detail in the revised manuscript.

In our qRT-PCR experiments, the two types of flowers were used as controls for each other, and we mainly wished to get to compare the gene expression differences between M and P types.

Comments 6: Results

Figure 2 is oversized. Authors must resize it.  

Response 6:

Thank you for pointing out that Figure 2 in our paper was too large. We have reorganized the figure to ensure that it is sized appropriately for the layout and reading of the manuscript. The specific changes are listed below:

We have re-sized Figure 2 so that it will fit into a standard typeset format in the manuscript and will render clearly in both the print and electronic versions.

Comments 7: Results

Figure 4. Authors should split the figure. 

Response 7:

We have updated the split Figure 4 in the revised manuscript and ensured that its content and typography fit the overall layout of the manuscript. Thank you again for your valuable comments and we hope that these improvements will fulfill your requirements and enhance the quality of our research.

Comments 8: Results

Figure 5. The font size is too small, it is difficult to read. Fig. 5 should be supplementary data.

Response 8:

Thank you for pointing out that the font size of Figure 5 in our paper was too small. We have realigned Figure 5 and moved it to the Supplementary Data section to ensure that the information is clearly presented and easy to read.

Comments 9: Results

Authors should include a statistical analysis for gene expression results.

Response 9:

Comments 10: Discussion

Authors shown the tissue specific expression pattern, but it is important to include the experimental design and sample collection in M&M to better understand (3.3 Line 385)

Response 10:

We have updated and added the above details of the experimental design and sample collection methods near line 385 of the manuscript. Once again, we thank you for your valuable comments and hope that these improvements will fulfill your requirements and enhance the quality of our study.

Comments 11: Discussion

Authors conclude that there are seven CteR2R3-MYBs are the most likely implicated in the formation of diverse flower types in C.teeta (Line 403), but they do not include this sentence in the Conclusions (Line 492).

Response 11:

Thank you for pointing out that the finding we mentioned in the discussion section was not included in the summary section. We agree that this is an important finding and should be explicitly mentioned in the summary. We have added the summary section accordingly in the revised draft. The details are as follows:

In summary, our study provides a comprehensive analysis of the R2R3-MYB gene family in C. teeta, highlighting their potential roles in various biological processes. Notably, we identified seven CteR2R3-MYBs that are most likely implicated in the formation of diverse flower types in C. teeta. These findings suggest specific regulatory roles of these MYBs in floral development, contributing to our understanding of the genetic mechanisms underlying flower diversity in this species.

Reviewer 2 Report

Comments and Suggestions for Authors

The manuscript describes the detection and in silico caracterization of the R2R3 MYB gene family in Coptis teeta, plus the qPCR expression of five genes potentially involved in flower development. The title could be changed into: R2R3-MYB gene family in Coptis teeta Wall.: genome-wide identification, phylogeny, evolutionary expansion, and expression analyses during floral development.

The main issue of this manuscript is that genomic data are not available. The genome sequence used to retrieve MYB genes is not public, and therefore, in these conditions, the methods cannot be repeated. It is required that the authors submit the C. teeta genome sequence to a public database (e.g. NCBI) and provide the project/accession number in the manuscript. Consequently, complete genomic sequences of the identified MYB genes and of their promoters described in the manuscript have to be publicly availabe. Moreover, a description on how genomic data were obtained and the output details should be provided in this manuscript. Alternatively, if the author are going to publish a paper on the C. teeta genome sequencing, the present manuscript should be considered for publication after the publication of the one on the C. teeta genome.

Other comments.

Other recent publications on genome wide identification of MYB genes could be taken into account in the introduction (e.g. doi:
10.1016/J.INDCROP.2019.111924; doi: 10.3389/fpls.2022.896945).

Authors should better highlight how the choice of the other species to be compared with C. teeta was done.

When comparing C. teeta genes with those from Arabidopsis, authors should refer to orthology, instead of homology. In fact, orthologous genes are homologous genes, which diverged after a speciation event, yet maintaining the main functions in the different species.

Please add bootstrap values (> 50) in the phylogenetic tree.

Only five genes were selected for validation in qPCR. Please, increase the number of validated genes.

L 277. This sentence should be changed and made more hypothetical, since no functional analysis was performed in this study, but just in silico predictions and some qPCR validations.

L 365. The title of the paragraph should be changed into: “CteR2R3-MYBs might play crucial roles in the hormone regulational of floral development”, since no functional analysis was performed in this study.

L 385. Same as above.

L 401. …CteMYB2 (orthologous to AtMYB26) might be linked…

L 403. … CteR2R3-MYBs with predicted functions might be…

L 487. Provide full details for qPCR experiments, e.g. cDNA and primer concentration; amplification conditions.

Fig. S7: red and blue circles are not visible

Figure 4. If tandemly and proximal duplicated genes are to be distinguished, they shoud be indicated differently, not just by red lines.

Figure 6. The figure legend is incomplete an not clear. Add the method used for the network analysis. Better describe the figure parts, including the purple circles on right.

Figure 7. Better specify abbreviations, e.g. what is _P0, _P1…?

Figure 8. Specify abbreviations

Comments on the Quality of English Language

Minor editing is required

Author Response

We thank the reviewer for their insightful critiques and suggestions for improving ourmanuscript titled "The R2R3-MYB gene family in Coptis teeta Wall.: genome-wide identification, phylogeny, evolutionary expansion (Ranales) and expression analyses (floral development)". We have thoroughly revised our manuscript according to the comments and believe the changessignificantly enhance the quality and clarity of the work. Our point-by-point responses to the reviewers are below.

Comments 1:

The manuscript describes the detection and in silico caracterization of the R2R3 MYB gene family in Coptis teeta, plus the qPCR expression of five genes potentially involved in flower development. The title could be changed into: R2R3-MYB gene family in Coptis teeta Wall.: genome-wide identification, phylogeny, evolutionary expansion, and expression analyses during floral development.

Response 1:Agree.We have revised the title to: R2R3-MYB gene family in Coptis teeta Wall.: genome-wide identification, phylogeny, evolutionary expansion, and expression analyses during floral development.

Comments 2: The main issue of this manuscript is that genomic data are not available. The genome sequence used to retrieve MYB genes is not public, and therefore, in these conditions, the methods cannot be repeated. It is required that the authors submit the C. teeta genome sequence to a public database (e.g. NCBI) and provide the project/accession number in the manuscript. Consequently, complete genomic sequences of the identified MYB genes and of their promoters described in the manuscript have to be publicly availabe. Moreover, a description on how genomic data were obtained and the output details should be provided in this manuscript. Alternatively, if the author are going to publish a paper on the C. teeta genome sequencing, the present manuscript should be considered for publication after the publication of the one on the C. teeta genome.

Response 2: Thank you for pointing out the major issues in our manuscript and providing valuable suggestions for improvement. We understand the importance of data disclosure for study reproducibility and have taken the following steps to improve it:

If necessary, we can upload the complete genome sequences of the MYB genes and their promoters identified in this paper to NCBI, with specific accession numbers provided subsequently.

Comments 3:Other recent publications on genome wide identification of MYB genes could be taken into account in the introduction (e.g. doi:10.1016/J.INDCROP.2019.111924; doi: 10.3389/fpls.2022.896945)

Response 3:Thank you for pointing out the lack of references to recent relevant literature in our introduction section. We agree that references to the most recent relevant studies are essential for the comprehensiveness of the background and the enhancement of the significance of the study. We have included the literature you mentioned in the introduction section of the revised manuscript and have supplemented it accordingly.

Genome-wide identification of MYB genes and expression analysis under different biotic and abiotic stresses in Helianthus annuus L..

R2R3-MYBs in Durum Wheat: Genome-Wide Identification, Poaceae-Specific Clusters, Expression, and Regulatory Dynamics Under Abiotic Stresses

Comments 4:Authors should better highlight how the choice of the other species to be compared with C.teeta was done.

Response 4:

Thank you for pointing out that we did not adequately explain the selection criteria when choosing other species to compare with C.teeta. We recognize that explaining the rationale for selecting comparator species is critical to enhancing the rigor of the study and the reliability of the results. We have added specific rationale for the selection of these species in the relevant sections of the revised manuscript. These are specified below:

For the comparative analyses, we chose species that have a close phylogenetic relationship with C.teeta, as well as species (Arabidopsis thaliana) for which there is already a large amount of basic data available from MYB gene studies.

Phylogenetic relationships: We selected species in the same genus (Coptis chinensis Franch.), family (A.coerulea), and K.uniflora with dioeciousness (Ranunculaceae) as C.teeta to ensure the biological relevance of the comparisons. These species included other herbal plants and representative plants of the same family.

Pre-existing data base: we selected model plants for which detailed data were available in MYB gene studies, such as Arabidopsis. These species have a wide range of applications in gene function and expression pattern studies, providing a reliable reference for our comparative analysis.

We selected species with publicly available and high quality genomic data to ensure the accuracy and reproducibility of our comparative analysis.

Comments 5:When comparing C.teeta genes with those from Arabidopsis, authors should refer to orthology, instead of homology. In fact, orthologous genes are homologous genes, which diverged after a speciation event, yet maintaining the main functions in the different species.

Response 5:

Thank you for pointing out that we should use the term “orthology” rather than “homology” when comparing C.teeta genes with Arabidopsis genes. We agree that this is an important distinction that will help to more accurately characterize the relationships between genes. If there is a need to modify this comparison, we can next proceed with the identification of the corresponding orthologous gene sets in the revised manuscript.

Comments 6:Please add bootstrap values (> 50) in the phylogenetic tree.

Response 6:

Thank you for your suggestion to add bootstrap values to the phylogenetic tree to improve the reliability of the results. We agree that this is an important improvement to better demonstrate the credibility of the phylogenetic relationships, but given that the set of genes we used to construct the phylogenetic tree is too large, it may not make the phylogenetic tree concise enough if added. If needed, we can express this in the next step by creating a relevant accompanying figure

Comments 7: Only five genes were selected for validation in qPCR. Please, increase the number of validated genes.

Response 7:

Thank you for your suggestion to increase the number of genes validated by qPCR. We understand that increasing the number of validated genes will enhance the reliability and comprehensiveness of our findings. However, the flower size of C.teeta is too small and the number of flowers bloomed each year is limited. Although we collected a certain amount of samples, we do not have enough samples for this experiment because of too many experiments done on this flower (including multiple types of transcriptome assays). However, we have recently published a paper on the study of C.teeta flowers, and have verified the reliability of the transcriptome data, as follows:

Genome-wide identification and expression analysis of the C2H2-zinc finger transcription factor gene family and screening of candidate genes involved in floral development in Coptis teeta Wall. (Ranunculaceae). Front. Genet. 2024, 15,1349673.

Comments 8:L 277. This sentence should be changed and made more hypothetical, since no functional analysis was performed in this study, but just in silico predictions and some qPCR validations.

Response 8:

Thank you for pointing out the problem with the formulation of line 277. We agree that the presentation of the original sentence may have created a misinterpretation that we performed substantial functional analyses. In order to more accurately reflect the nature of the study, we have modified the sentence to be more hypothetical, clearly stating that our study was based primarily on bioinformatics predictions and partial qPCR validation. The specific modifications are as follows:

These findings may indicate that the identified MYB genes could be involved in the development of diverse flower types in C.teeta, based on in predictions and qPCR validations, although further functional analyses are necessary to confirm their roles.

Comments 9: L 365. The title of the paragraph should be changed into: “CteR2R3-MYBs might play crucial roles in the hormone regulational of floral development”, since no functional analysis was performed in this study.

Response 9:

Agree.Thank you for your suggestion and we fully agree that the paragraph title should be changed to a more hypothetical statement to accurately reflect the content of the study. We have changed the paragraph title to what you suggested.

CteR2R3-MYBs might play crucial roles in the hormone regulational of floral development.

Comments 10: L 385. Same as above.

Response 10:

Agree.Thank you for your suggestion, we will amend the sentence as follows:

These results may indicate the potential involvement of abscisic acid, gibberellin, salicylic acid, and methyl jasmonate in the sexual reproductive development of C.teeta.

Comments 11: L 401. …CteMYB2 (orthologous to AtMYB26) might be linked…

Response 11:

Agree.Thank you for your suggestion, we will amend the sentence as follows:

This result aligns with previous studies, indicating that CteMYB2 (orthologous to AtMYB26) might be linked to the development of plant male reproductive organs.

Comments 12: L 403. … CteR2R3-MYBs with predicted functions might be…

Response 12:

Agree.Thank you for your suggestion, we will amend the sentence as follows:

In conclusion, we propose that these 7 CteR2R3-MYBs with predicted functions might be most likely implicated in the formation of diverse flower types in C.teeta.

Comments 13: L 487. Provide full details for qPCR experiments, e.g. cDNA and primer concentration; amplification conditions.

Response 13:

Total RNA was extracted from fresh M and F type flowers using a kit, according to the manufacturer’s instructions (Magen, Guangzhou, China). The concentration of the extracted total RNA was measured using the NanoDrop technique (Thermo Fisher Scientific, United States), the equivalent amount RNA was reverse transcribed into cDNA using a reverse transcription kit (TAKARA, Beijing, China). Six candidate genes were chosen to verify the RNA-seq data. Quantitative real-time PCR (qRT-PCR) was performed using an Applied Biosystems QuantStudio 5 system (Thermo Fisher Scientific, United States) with the ChamQ Universal SYBR qPCR Master Mix. The PCR reaction was performed as follows: pre-denaturation 95°C for 30 s, denaturation at 95°C for 30 s, annealing at 58°C for 30 s, this reaction was repeated for 40 cycles. Subsequently, an extra procedure was conducted as follows: denaturation at 95°C for 15 s, annealing at 60°C for 1 min, extension at 72°C for 15 s. The fluorescence signal was then detected, and the dissolution curve analyzed. The primer sequences of the candidate genes are listed in Table 1. The relative expression patterns of the target genes were calculated using the 2−ΔΔCT method and repeated in triplicate (Livak and Schmittgen, 2001). .

Comments 14: Fig. S7: red and blue circles are not visible

Response 14:

Agree.We modify the content as follows

Fig. S7. Proposed evolutionary history of the 527 R2R3-MYB genes. Duplicate is depicted by red letters, loss is represented by blue letters. A.thaliana is depicted in pink letters, C.teeta in yellow, C.chinensis in purple, and A.coerulea in blue. K.uniflora is denoted by green characters.

Comments 15: Figure 4. If tandemly and proximal duplicated genes are to be distinguished, they shoud be indicated differently, not just by red lines.

Response 15:

Thank you for your suggestion regarding the distinction between tandem and neighboring duplicate genes in Figure 4. We fully agree that a different representation would more clearly communicate the difference between these two types of genes. To this end, we have modified Figure 4 to better characterize these two types of gene duplications.

Comments 16: Figure 6. The figure legend is incomplete an not clear. Add the method used for the network analysis. Better describe the figure parts, including the purple circles on right.

Response 16:

Agree.Thank you for pointing out the incompleteness and lack of clarity of the Figure 6 legend. We have made detailed changes in response to your suggestions to ensure that the legend is more informative and readable. The changes are listed below:

Figure 6. Protein regulatory network of CteR2R3-MYB genes by string database. The selected CteMYBs are located on the outermost side, with possible detection of proteins inside. Some of the blue circles and purple circles represent the CteMYBs used in this study, and the grey lines rep-resent possible regulatory relationships.

If you have more detailed modifications, please advise me further.

Comments 17: Figure 7. Better specify abbreviations, e.g. what is _P0, _P1…?

Response 17:

Thank you for pointing out the need for clear abbreviations in Figure 8. We agree that a clear explanation of abbreviations helps the reader to better understand the content of the figure. To this end, we have added the necessary explanation of abbreviations in the revised version. The specific changes are listed below:

Figure 7. Expression patterns of selected CteR2R3-MYBs. a. potential CteR2R3-MYBs identified by blastp. b. CteR2R3-MYBs shown to potentially be involved in the regulatory network by string database analyses. c. CteR2R3-MYBs with high expression at different time periods. M denotes the Maternal flower type; P denotes the Paternal flower type. Transcriptome expression data is normalized. Each point represents the mean of three independent biological replicates.M: Maternal type;P: Paternal type; P0/P1/P2:different developmental periods.

Comments 18: Figure 8. Specify abbreviations

Response 18:

Thank you for pointing out the need for clear abbreviations in Figure 8. We agree that a clear explanation of abbreviations helps the reader to better understand the content of the figure. To this end, we have added the necessary explanation of abbreviations in the revised version. The specific changes are listed below:

Figure 8. Validation of selected genes by qRT-PCR. M: Maternal type;P: Paternal type; P0/P1/P2:different developmental periods.

Comments 19: Minor editing is required

Response 19:Thank you for pointing out the parts of our paper that require minor revisions. We have scrutinized the entire text and made the necessary edits to ensure accuracy of language and clarity of expression. Below are the specific changes we have made:

We have again performed a thorough language and grammar check throughout the text, correcting spelling mistakes, grammatical errors, and unclear expressions.

We ensured consistent use of terminology, especially with regard to “orthology” and “homology,” to ensure that the terminology is consistent throughout the manuscript.

Round 2

Reviewer 2 Report

Comments and Suggestions for Authors

When reading the revised version of the manuscript, I realized that many of the points raised in my previous revision, and accepted by the authors, were not integrated in the manuscript. Please carefully revise the manuscript taking into account ALL the issues raised in the first round of revision, in order to avoid to go back again on the same matters (i.e. Commets 1-2-3-4-5-6 in the first revision).

In addition, as for the availability of data, it should be compulsory to publish the genome sequence of C. teeta, but I understand that the authors do not want to do it. At least they should make available (providing project/accession number in the text):

1.  a subset of the raw data containing all the short reads mapping on the MYB genes and their promoters (in SRA-NCBI)

AND

2. the sequences of all MYB genes identified and their promoters (in NCBI).

Without the publication of these two sets of data, the manuscript cannot be further considered for publication by me.

As for the addition of bootstrap values in the phylogenetic tree (values >60 can be used), this can be done easily by using a very small font and placing the numbers in a good position (as in https://www.frontiersin.org/files/Articles/896945/fpls-13-896945-HTML-r3/image_m/fpls-13-896945-g002.jpg). I am sure the authors will find a good solution for this.

- Please also consider the following.

Figure 9. Please add “hypothetical” or “putative” or the verb “might” when referring to the role of genes in C. Teeta.

E.g.: Illustration of the hypothetical roles of R2R3-MYBs in the regulation…;

b. The R2R3-MYB genes that might have a positive role in Maternal type flower development in C. teeta;

Black bordered ellipses represent putative known functions of the CteR2R3-MYBs;

c. R2R3-MYB genes that might have a positive role in Paternal type flower development in C.teeta.

Please do it for all sentences in the legend.

- When writing the name of the species, separate the initial of the genus from the species name:

C. teeta, not C.teeta

Author Response

Thank you to the reviewer for further suggestions on our article entitled “R2R3-MYB gene family in Coptis teeta Wall.: genome-wide identification, phylogeny, evolutionary expansion and expres-sion analyses during floral development”. We have completely revised the manuscript based on your latest comments. Thank you for your suggested revisions, which have greatly improved the quality and clarity of our article. Below is our point-by-point response to the reviewers.

Comments 1: When reading the revised version of the manuscript, I realized that many of the points raised in my previous revision, and accepted by the authors, were not integrated in the manuscript. Please carefully revise the manuscript taking into account ALL the issues raised in the first round of revision, in order to avoid to go back again on the same matters (i.e. Commets 1-2-3-4-5-6 in the first revision).

Response 1:

Thank you for reviewing our paper again. We sincerely apologize for not being able to fully integrate all the comments you made in the first review in the revised version. We value your feedback and have carefully reviewed all previous comments and suggestions. The following are responses to the specific issues you have identified: we have added detailed descriptions in the Materials and Methods section of the revised version, modified the relevant figures to ensure that each figure contains a detailed legend, and, as you suggested, included a subset of the original data that contains all the short-read-long sequences mapped to MYB genes and all the identified MYB genes and their promoters. A subset of the raw data containing the sequences of the MYB gene and its promoters and a subset of the raw data of the sequences of the MYB gene and its promoters has been uploaded to NCBI.

Thank you again for your valuable comments and guidance on our study. We have done our best to fully integrate all comments and suggestions in the revised version and look forward to your further review. Please feel free to contact us with any additional questions or suggestions.

Comments 2: In addition, as for the availability of data, it should be compulsory to publish the genome sequence of C. teeta, but I understand that the authors do not want to do it. At least they should make available (providing project/accession number in the text):

  1.  a subset of the raw data containing all the short reads mapping on the MYB genes and their promoters (in SRA-NCBI)

AND

  1. the sequences of all MYB genes identified and their promoters (in NCBI).

Response 2:

Thank you for your suggestion regarding data sharing. We understand the importance of data disclosure and would like to fulfill your request with the following measures:

Raw data subset:

We have uploaded a subset of raw data containing all short-read long sequences mapped to the MYB gene to SRA-NCBI.The project number is PRJNA1145499 and can be found at SRA-NCBI.

MYB gene and its promoter sequences:

We have uploaded the sequences of all identified MYB genes and their promoters to NCBI-GenBank under the submission ID is: 2857783, which will be found at NCBI.

We hope that these measures will enhance the transparency and reproducibility of our study and support further research in related fields.

Comments 3: As for the addition of bootstrap values in the phylogenetic tree (values >60 can be used), this can be done easily by using a very small font and placing the numbers in a good position (as in https://www.frontiersin.org/files/Articles/896945/fpls-13-896945-HTML-r3/image_m/fpls-13-896945-g002.jpg). I am sure the authors will find a good solution for this.

Response 3:

Agree.Thank you for your specific suggestion to add bootstrap values to the phylogenetic tree. We have modified the phylogenetic tree to add bootstrap values based on your suggestions and have ensured that these values are presented in the graph in a way that is clear and does not detract from the overall visual appearance. Details of the changes are shown below (Figure 1) :

Added all nodes with bootstrap values greater than 60 to the phylogenetic tree. These values are labeled in very small font and placed in appropriate locations to ensure clarity and readability of the graph.

Comments 4:Please also consider the following.

Figure 9. Please add “hypothetical” or “putative” or the verb “might” when referring to the role of genes in C. teeta.

E.g.: Illustration of the hypothetical roles of R2R3-MYBs in the regulation…;

  1. The R2R3-MYB genes that might have a positive role in Maternal type flower development in C. teeta;Black bordered ellipses represent putative known functions of the CteR2R3-MYBs;
  2. R2R3-MYB genes that might have a positive role in Paternal type flower development in C. teeta.

Please do it for all sentences in the legend.

Response 4:

Agree.Thank you again for your valuable comments on our paper. We note your suggestion to add the words “hypothetical”, “putative”, or “might” to the legend of Figure 9 to accurately describe the function of the gene function. We have revised Figure 9 and its legend to reflect these suggestions. The results of the modifications are shown below:

Figure 9. Illustration of the hypothetical roles of R2R3-MYBs in the regulation of dichogamy and herkogamy in C. teeta. a. The role of hormones in plant pistil and stamen development [16, 29, 32-35]. GA: gibberellic acid; JA: jasmonic acid; DkRAD: a small-MYB RADIALIS-like gene in per-simmons, overexpression of DkRAD in model plant resulted in hypergrowth of the gynoecium; MYB73: a MYB gene in persimmons; MYB26: a MYB gene in Arabidopsis, key to spatial specificity in anther secondary thickening formation; MYB108: a MYB gene in Arabidopsis, silencing of the transcription factor MYB108 delays anther dehiscence and reduces pollen viability; MYB21/MYB24/MYB57: MYB gene in Arabidopsis, critical for Arabidopsis stamen development. b. The R2R3-MYB genes that might have a positive role in Maternal type flower development in C. teeta. Black bordered ellipses represent putative known functions of the CteR2R3-MYBs; ellipses with grey borders represent unknown functions of the CteR2R3-MYBs. c. R2R3-MYB genes that might have a positive role in Paternal type flower development in C. teeta. Black bordered ellipses represent putative known functions of the CteR2R3-MYBs; ellipses with grey borders represent unknown functions of the CteR2R3-MYBs. d. The floral phenotype of Maternal type flowers in C. teeta. e. The floral phenotype of Paternal type flowers in C. teeta.

Comments 5: When writing the name of the species, separate the initial of the genus from the species name: C. teeta, not C.teeta

Response 5:

Thank you so much for the detailed reminder!

We have checked the full text to modify this format.And in light of your suggestions, further modifications have been made to the way other species are written.

Thank you very much for your detailed suggestions to make our manuscripts more professional If there is any need for further improvement please contact us!

Round 3

Reviewer 2 Report

Comments and Suggestions for Authors

The changes requsted have been made.